# Low Expression of the *NRP1* Gene Is Associated with Shorter Overall Survival in Patients with Sonic Hedgehog and Group 3 Medulloblastoma

**DOI:** 10.3390/ijms241411601

**Published:** 2023-07-18

**Authors:** Moisés Augusto de Araújo, Osvaldo Malafaia, Jurandir M. Ribas Filho, Livia Fratini, Rafael Roesler, Gustavo R. Isolan

**Affiliations:** 1Graduate Program in Principles of Surgery, Mackenzie Evangelical University, Curitiba 80730-000, PR, Brazil; 2The Center for Advanced Neurology and Neurosurgery (CEANNE), Porto Alegre 90560-010, RS, Brazil; 3Department of Pharmacology, Institute for Basic Health Sciences, Federal University of Rio Grande do Sul, Porto Alegre 90035-003, RS, Brazil; 4Cancer and Neurobiology Laboratory, Experimental Research Center, Clinical Hospital (CPE-HCPA), Federal University of Rio Grande do Sul, Porto Alegre 90035-003, RS, Brazil; 5National Science and Technology Institute for Children’s Cancer Biology and Pediatric Oncology–INCT BioOncoPed, Porto Alegre 90035-003, RS, Brazil; 6Research Center, Moinhos de Vento Hospital, Porto Alegre 90035-001, RS, Brazil; 7Spalt Therapeutics, Porto Alegre 90560-010, RS, Brazil

**Keywords:** neuropilin-1, NRP1, medulloblastoma, pediatric cancer, brain tumor

## Abstract

Medulloblastoma (MB) is the most common type of malignant pediatric brain tumor. Neuropilin-1 (NRP1), encoded by the NRP1 gene, is a transmembrane glycoprotein overexpressed in several types of cancer. Previous studies indicate that NRP1 inhibition displays antitumor effects in MB models and higher NRP1 levels are associated with poorer prognosis in MB patients. Here, we used a large MB tumor dataset to examine NRP1 gene expression in different molecular subgroups and subtypes of MB. We found overall widespread NRP1 expression across MB samples. Tumors in the sonic hedgehog (SHH) subgroup showed significantly higher NRP1 transcript levels in comparison with Group 3 and Group 4 tumors, with SHH samples belonging to the α, β, Δ, and γ subtypes. When all MB subgroups were combined, lower NRP1 expression was associated with significantly shorter patient overall survival (OS). Further analysis showed that low NRP1 was related to poorer OS, specifically in MB subgroups SHH and Group 3 MB. Our findings indicate that patients with SHH and Group 3 tumors that show lower expression of NRP1 in MB have a worse prognosis, which highlights the need for subgroup-specific investigation of the NRP1 role in MB.

## 1. Introduction

Medulloblastoma (MB) is the most frequent type of malignant brain tumor in children and is an important cause of cancer-related morbidity and mortality in pediatric patients. Multimodal treatment with chemotherapy, radiotherapy, and surgery has improved cure rates, but about one-third of patients still relapse, and survivors experience long-term neurological, cognitive, and endocrinological sequalae [1,2]. Classification into molecular subgroups has greatly contributed to the advancement of our understanding of MB biology and clinical prognosis. The four consensus molecular subgroups of MB are wingless-activated (WNT), sonic hedgehog (SHH), Group 3, and Group 4; patients bearing Group 3 and Group 4 tumors having a particularly poor prognosis [2,3,4,5]. Within each subgroup, intra- and intertumoral heterogeneity has led to further classification into twelve subtypes [6,7]. MB arises in the cerebellum, with different molecular subgroups originating from diverse cells of origin [8,9].

MB hijacks the biological mechanisms that mediate normal central nervous system (CNS) development and plasticity [10,11,12]. Neuropilin-1 (NRP1), a transmembrane glycoprotein encoded by the NRP1 gene, is found in vertebrates and plays a role in neuronal development, notably by guiding axons through a mechanism dependent on semaphorine (SEMA) proteins, particularly Class 3 SEMA [13]. NRP1 also influences cell signaling and cell function by acting as a co-receptor for vascular endothelial growth factor (VEGF) and its receptor VEGFR, in addition to interacting with other growth factors including placental growth factor (PlGF), fibroblast growth factor (FGF), hepatocyte growth factor (HGF), platelet-derived growth factor (PDGF), and transforming growth factor β (TGF- β) [14].

NRP1 expression is abnormally increased in various cancer types, including MB [14,15,16,17]. Overexpression of NRP1 has been associated with worse prognosis in lung cancer [18], pancreatic cancer [19], liver cancer [16], breast cancer [20], and glioma [15]. One previous study indicated that high NRP1 levels are correlated with poor prognosis in MB patients [21] and that NRP1 inhibition resulted in antitumoral effects in experimental models of MB [17,21,22,23]. Given that pediatric brain tumors likely arise from abnormalities in CNS development during embryogenesis, they may be particularly sensitive to factors that influence neurodevelopment, such as NRP1. Here, we analyzed NRP1 transcript levels and their possible association with overall survival (OS) in different molecular subgroups and subtypes of MB tumors.

## 2. Results

### 2.1. NRP1 Transcript Levels in Different MB Molecular Subgroups

There was widespread *NRP1* expression across MB tumors. Tumors in the SHH subgroup (*n* = 223) showed significantly higher transcript levels of *NRP1* in comparison with Group 3 (*n* = 144) and Group 4 (*n* = 326) tumors (*p*s < 0.001; Figure 1).

### 2.2. NRP1 Transcript Levels in Different MB Subtypes

MB tumors belonging to the WNT β, SHH γ, Group 3 α, Group 4 α, and Group 4 γ subtypes showed apparent lower levels of *NRP1* expression compared to other subtypes, particularly WNT α (Figure 2). 

### 2.3. Lower NRP1 Expression Is Associated with Shorter OS in Patients with MB

Analysis of MB patient OS in relation to *NRP1* tumor transcript levels revealed that, in the set of tumors combining all MB subgroups (*n* = 612), lower *NRP1* expression was associated with shorter OS (*p* < 0.0001). When the molecular subgroups were analyzed separately, we observed that low *NRP1* was related to shorter patient OS specifically in MB subgroups SHH and Group 3 (both *p*s < 0.05). In contrast, patients with Group 4 tumors showed an apparent reduction in OS when NRP1 levels were higher, although this effect did not reach statistical significance (Figure 3).

### 2.4. NRP1 Transcript Levels in MB Molecular Subgroups in a Second Dataset

Analysis of data in another, smaller MB dataset, namely the Tumor Medulloblastoma–Pfister (*n* = 67), originally described in reference [5], confirmed significantly lower *NRP1* transcription levels in Group 3 (*n* = 41, *p* < 0.001) and Group 4 (*n* = 64, *p* < 0.0001) tumors compared with SHH tumors (*n* = 46, Figure 4).

## 3. Discussion

Previous evidence has indicated that NRP1 inhibition displays antitumor effects in experimental MB. Knockdown of NRP1 reduced the growth of orthotopic D283 and D341 MB xenografts and prevented spinal metastasis in mice, resulting in significantly prolonged survival, without affecting MB cell proliferation per se. In addition, blocking NRP1 with specific antibodies prevented PlGF-induced activation of protein kinase pathways in human and murine MB cells [17]. NRP1 inhibition with the peptidomimetic agent MR438 reduced self-renewal capacity, invasiveness, and stemness markers in MB stem cells obtained from MB cell line cultures [22]. MR438 also sensitized MB stem cells to radiotherapy in vitro and improved the efficacy of radiotherapy in vivo in a heterotopic xenograft mouse model of MB [23]. Transient expression of microRNA MiR-148a in non-WNT medulloblastoma cell lines resulted in impaired proliferation, survival, invasiveness, and tumorigenicity of MB cells associated with a reduction in NRP1 expression, whereas restoration of NRP1 rescued the disruption of invasion potential and tumorigenicity [21]. Together, these findings indicate that NRP1 inhibition displays antitumor effects on MB by influencing cell survival, invasiveness, and stemness. On the basis of evidence for a stimulating role of the PlGF/NRP1 pathway in pediatric cancers, a recent open-label Phase I clinical trial aimed to evaluate TB-403, a monoclonal antibody against PlGF, in pediatric patients with relapsed or refractory MB, neuroblastoma, Ewing sarcoma, or alveolar rhabdomyosarcoma [25].

Immunohistochemical analyses found NRP1 overexpression in MB (*n* = 5 samples) compared to non-tumoral pediatric cerebellar tissue (*n* = 2). Additionally, over 90% of a set of 32 MB samples of different molecular subtypes showed strong NRP1 expression [17]. NRP1 levels were also previously studied by immunohistochemistry in 93 formalin fixed paraffin-embedded MB tumor sample tissues. Around 75% of WNT subgroup tumors showed no detectable NRP1, whereas 23% of Group 3 tumors lacked NRP1 expression. Patients bearing tumors with moderate or high NRP1 levels had significantly shorter overall survival than those with no detectable, or low, NRP1 expression [21]. Another analysis of a small set of MB samples (34 samples with low NRP1 expression and 8 samples with high NRP1 expression) found reduced survival in patients with high-expressing MB [17]. Collectively, these data suggest that one could expect high NRP1 levels to be associated with a poorer prognosis.

The present findings, obtained from a larger patient cohort, indicate that WNT and SHH tumors, which present relatively better prognosis, show higher *NRP1* expression compared to Group 3 and Group 4 MB. This pattern was further confirmed in a second MB tumor dataset. In addition, there is a marked difference between subtypes (α or β) within the Wnt subgroup, as well as higher expression in α and β subtypes in comparison with γ tumors within the Group 4 molecular subgroup. Unexpectedly, our present results indicate that MB patients with SHH and Group 3 tumors that display lower levels of *NRP1* transcription have a worst prognosis. A similar pattern was observed in Wnt tumors, although the effect did not reach significance, likely because of a smaller number of samples. In contrast, patients with Group 4 tumors with high *NRP1* show an apparent decline in OS. It is possible that the heterogeneity in *NRP1* levels between subtypes within specific subgroups (such as Wnt) impacts the association of transcript levels and OS when all tumors belonging to a given subgroup are analyzed together. It should be noted that *NRP1* expression may be a minor contributing factor to the already poorer outcome of patients with Group 3 and Group 4 tumors when compared to other molecular subgroups.

Although most previously published findings indicate that NRP1 promotes tumor growth and contributes to reducing cancer patient survival [17,21], it might not always be the case. For instance, the NRP1 ligand PIGF acts as a pro-tumoral factor in some models but can inhibit tumor growth in others [26]. Moreover, higher NRP1 expression was reported to be associated with a longer survival time in patients with neuroblastoma (NB), a pediatric tumor usually arising from the peripheral nervous system. The same study showed that NRP1 knockdown promoted migration and invasion in NB cells, suggesting a tumor suppressive role for NRP1, via activation of β1 integrin [27]. Together with the present results, these findings raise the possibility that the role of NRP1 in MB in different tumor subtypes is more complex than previously thought and highlight the need for further experimental studies that explore the role of NRP1 in a subgroup-specific manner.

## 4. Materials and Methods

### 4.1. Gene Expression, Tumor and Patient Data

Analyses of NRP1 transcription were first performed in a previously described transcriptome dataset comprising 763 tumor samples from patients with MB (Cavalli cohort, (GEO: GSE85218)) [24]. Expression levels were normalized using the R2 Genomics Analysis and Visualization Platform (http://r2.amc.nl). Tumors were classified into different molecular subgroups and subtypes according to data available in the dataset. A feature of the R2 platform, namely the Kaplan Scan (KaplanScan algorithm), where an optimum survival cut-off is established based on statistical testing, was used.

The database Tumor Medulloblastoma-Pfister-167 [5] was downloaded from R2: Genomics Analysis and Visualization Platform (https://hgserver1.amc.nl/cgi-bin/r2/main.cgi (accessed on 12 July 2023)). Data were acquired, normalized, and log2 transformed. The number of samples analyzed per molecular subgroup was WNT, *n* = 16, SHH, *n* = 46, Group 3, *n* = 41, and Group 4, *n* = 64 MB tumors. The “ggplot2”, “ggsignif”, and “tidyverse” packages were used for analysis and graph plotting. The “PMCMRplus” package was used for statistical analysis. The packages were used in R version 4.3.0.

### 4.2. Statistical Analysis

Data are presented in box plot format as log2-transformed signal intensity. Comparisons in transcript levels were performed using the Welch’s ANOVA through the R2 platform, with *p* values < 0.01 considered to indicate significant statistical differences. Results are presented in boxplot format as log2-transformed signal intensity, with data shown as median and whiskers, minimum to maximum. Patient OS was measured from the day of diagnosis until death or the date of last follow-up. OS was calculated using the Kaplan–Meier estimate, with median values and long-rank statistics; *p* < 0.05 would indicate significant differences between groups in OS. Data from the Pfister cohort were analyzed with the Kruskal–Wallis analyses if variance followed by Dunn’s tests.

## 5. Conclusions

In summary, the present study is the first to report a subgroup-specific decline in OS associated with a reduction in NRP1 expression in MB tumors. This finding highlights the importance of identifying differences among MB groups when characterizing biomarkers and therapeutic targets. Further experiments should aim to clarify whether NRP1 has differential effects on MB growth depending on molecular subtype.

## Figures and Tables

**Figure 1 ijms-24-11601-f001:**
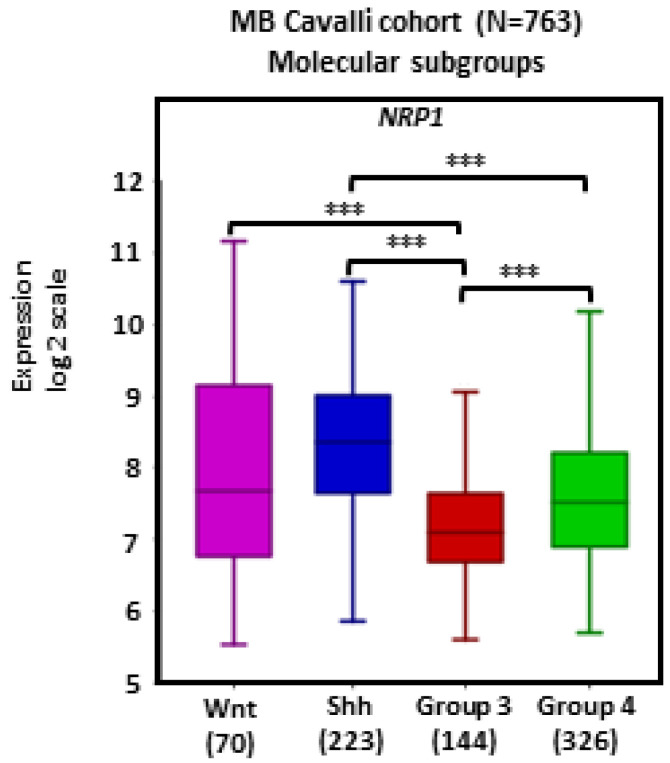
*NRP1* gene expression in different molecular subgroups of human MB. Tumors from the dataset, as described by Cavalli et al. [24], were analyzed with the R2 Genomics Analysis and Visualization Platform (http://r2.amc.nl). Results are presented in boxplot format as log2-transformed signal intensity. Bars show data for Group 3 (*n* = 144), Group 4 (*n* = 326), SHH (*n* = 223), and WNT (*n* = 70) MB; *** *p* < 0.001.

**Figure 2 ijms-24-11601-f002:**
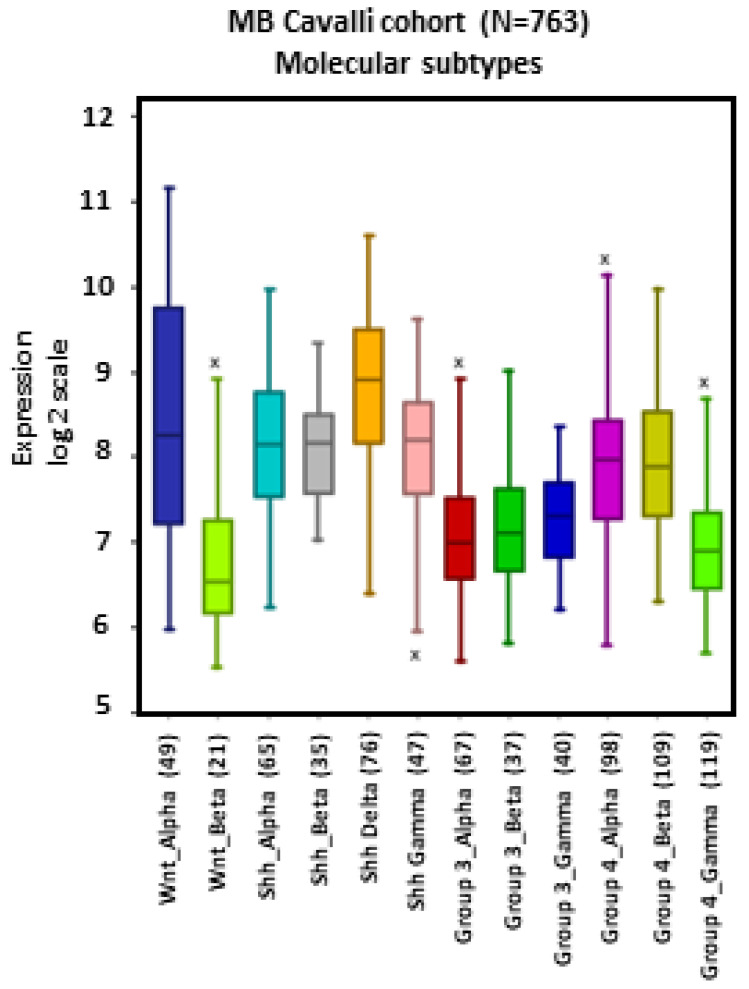
*NRP1* gene expression across different subtypes of human MB. Tumors from the dataset, described by Cavalli et al. [24], were analyzed with the R2 Genomics Analysis and Visualization Platform (http://r2.amc.nl). Results are presented in boxplot format as log2-transformed signal intensity. Bars show data for all 12 MB subtypes.

**Figure 3 ijms-24-11601-f003:**
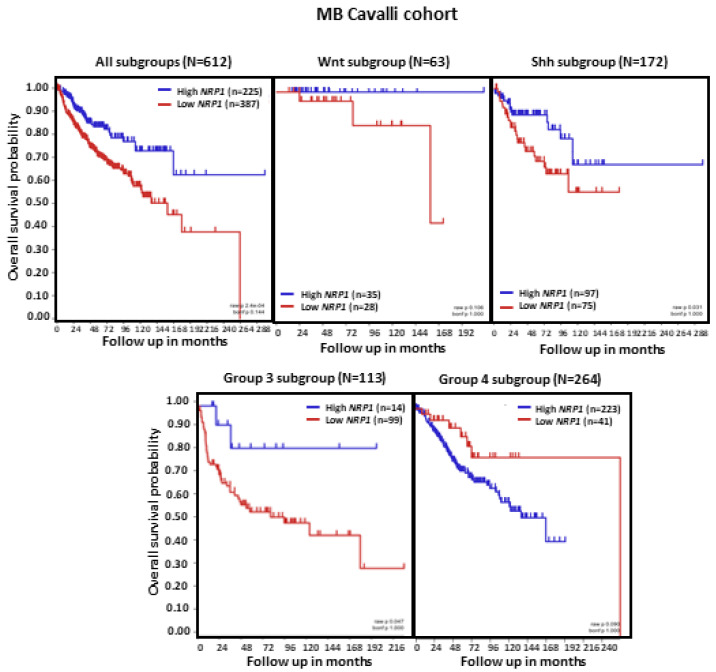
*NRP1* expression and OS in patients with MB. Results are shown for all MB tumors combined and separately according to molecular subgroup. Tumors from the dataset, described by Cavalli et al. [24], were analyzed. Patient OS was measured from the day of diagnosis until death or the date of last follow-up and calculated using the Kaplan–Meier estimate, with median values and long-rank statistics; *p* < 0.05.

**Figure 4 ijms-24-11601-f004:**
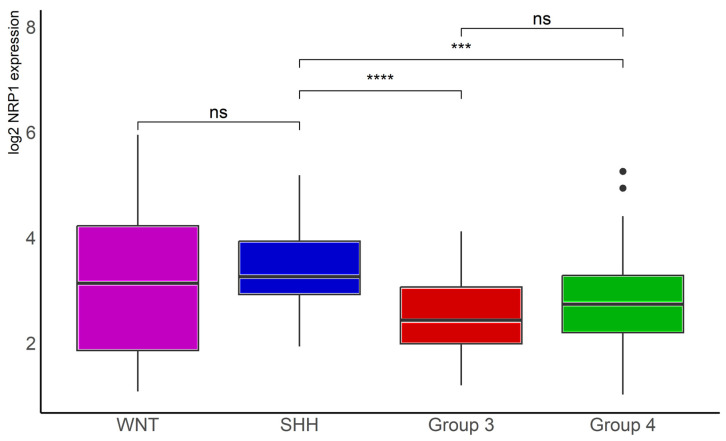
*NRP1* gene expression in different molecular subgroups of human MB. Data from the Tumor Medulloblastoma–Pfister (*n* = 67) dataset [5] were obtained through R2: Genomics Analysis and Visualization Platform (https://hgserver1.amc.nl/cgi-bin/r2/main.cgi (accessed on 12 July 2023)), normalized, and log2 transformed. The “ggplot2”, “ggsignif”, and “tidyverse” packages were used for analysis and graph plotting. WNT, *n* = 16, SHH, *n* = 46, Group 3, *n* = 41, and Group 4, *n* = 64; *** *p* < 0.001 and **** *p* < 0.0001.

## Data Availability

The data presented in this study are available upon request from the corresponding author.

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
