# Peer review of "Low Expression of the NRP1 Gene Is Associated with Shorter Overall Survival in Patients with Sonic Hedgehog and Group 3 Medulloblastoma"

_ijms, 2023, doi:10.3390/ijms241411601_

Round 1

Reviewer 1 Report

While the results reported in this manuscript seem interesting, there are some pitfalls which make the conclusion unconvincing.

 NRP1 is a transmembrane receptor, its function in promoting caner proliferation and survival relies upon the ligands PIGF. Including the expression information of PIGF will makes the results meaningful.

The prognosis of the fourth subgroup MB varies, group 3 has the worst prognosis, with relative lower NRP1 expression. The overall survival throughout all the groups, the lower survival/low expression were mainly attributed to group 3. NRP1 may be just a minor contributing factors.

The low/high expression cutoff (%) utilized among all the groups are completely different. For example, in group 4, the high/low cutoff was % 85/15, while in group 3, the high/low cutoff was % 12/88. This seems not a fair comparison, and the conclusion might be very misleading.

Author Response

Manuscript number: ijms-2450151-R1

RESPONSE TO REVIEWER 1:

  1. Reviewer’s comment: NRP1 is a transmembrane receptor, its function in promoting caner proliferation and survival relies upon the ligands PIGF. Including the expression information of PIGF will makes the results meaningful.

RESPONSE: This is a very good suggestion. However, NRP1 acts by serving as co-receptor for various other ligands, including vascular endothelial growth factor (VEGF), its receptor VEGFR, fibroblast growth factor (FGF), hepatocyte growth factor (HGF), platelet-derived growth factor (PDGF), and transforming growth factor β (TGF- β). Thus, there is no reason to evaluate the expression of PIGF alone. Future studies can focus on NRP1 ligands rather than NRP1, which is the focus of the current study.

  1. Reviewer’s comment: The prognosis of the fourth subgroup MB varies, group 3 has the worst prognosis, with relative lower NRP1 expression. The overall survival throughout all the groups, the lower survival/low expression were mainly attributed to group 3. NRP1 may be just a minor contributing factors.

RESPONSE: Even if patients with Group 3 tumors already show lower survival rates regardless of NRP1 expression levels, when one looks at our data specifically from Group 3 tumors, it is clear that low NRP1 expressions contributes significantly to make overall expression even lower. We have included an additional comment in the Discussion section to address the comment made by the Reviewer.

  1. Reviewer’s comment: The low/high expression cutoff (%) utilized among all the groups are completely different. For example, in group 4, the high/low cutoff was % 85/15, while in group 3, the high/low cutoff was % 12/88. This seems not a fair comparison, and the conclusion might be very misleading.

RESPONSE: We conducted the data analysis with a widely used feature of the R2 platform, which is the Kaplan Scan (KaplanScan algorithm), where an optimum survival cut-off is established based on statistical testing instead of, for example, just choosing the mean value or median. The Kaplan scanner separates the samples of a dataset into two groups based on the gene expression of one gene. In the order of expression, it will use every increasing expression value as a cutoff to create 2 groups and test the p-value in a log rank test. The highest value is then reported, accompanied by a Kaplan Meier picture. So in short, it will find the most significant expression cut off for survival analysis. The best possible Kaplan Meier curve is based on the log rank test, which is the data transformation in which numerical or ordinal values are replaced by their rank when the data are sorted by expression. This transformation is useful for non-parametric statistical tests. Moreover, the number of sample patients may vary in the groups due to availability of data survival in the dataset. We have included a comment in the Methods section of the article in order to further clarify this point.

Reviewer 2 Report

The authors present data from in silico analysis of NRP1 in Medulloblastoma from publicly available dataset and conclude that low expression of the NRP1 gene is associated with shorter survival in patients with SHH and Group 3 Medulloblastoma. It is a very simple paper. Researchers just used the Cavalli patient data sets and stratified prognosis based on NRP1 expression without any further validation studies. 

All the figures look pixelated and could be submitted with high resolution.

Figure 1: NRP1 gene expression in different molecular subgroups of human MB 

o   Would be better to put the groups in order (WNT, SHH, Group 3, Group 4) from left to right

o   Any comments on WNT subtype? Seems like it has high expression of NRP1 as well 

o   Says for values to be statistically significant, then p<0.001. Methods says p<0.01 

Figure 2: NRP1 gene expression across different subtypes of human MB 

o   Some Group 4 and WNT subtypes also seem to have higher expression levels of NRP1 – would be helpful if researchers commented on this as well  

Figure 3:

o   Paper comments on SHH, G3, and G4 patients. Should comment on WNT as well as there is a figure panel for it 

Other comments: 

-       Researchers can go a step further and try to use other patient data sets to validate their findings 

-       Perhaps suggest a reason for why SHH and G3 tumours with low NRP1 expression, have poor prognosis when the literature says otherwise (high NRP1 = worse prognosis in multiple cancer types) 

-       Must improve figures. They are blurry and not easy to read

-       What is the rationale for choosing NRP1? How did the researchers come about finding NRP1 as a gene target (apart from it being overexpressed in other cancers)? Why did they decide to look at it in specific subtypes?

-       In the discussion, authors wrote “MRP1” instead of NRP1 

-       The authors have mentioned in methods that “MicroRNA ex-153 pression levels were normalized using the R2 Genomics Analysis and Visualization Plat-154 form (http://r2.amc.nl).” Where is this data displayed? 

Although the finding that low expression of the NRP1 is associated with shorter survival in patients with SHH and Group 3 Medulloblastoma seems noteworthy, interest to the readers may be dampened without any experimental validation studies. 

Author Response

Manuscript number: ijms-2450151-R1

RESPONSE TO REVIEWER 2:

  1. Reviewer’s comment: The authors present data from in silico analysis of NRP1 in Medulloblastoma from publicly available dataset and conclude that low expression of the NRP1 gene is associated with shorter survival in patients with SHH and Group 3 Medulloblastoma. It is a very simple paper. Researchers just used the Cavalli patient data sets and stratified prognosis based on NRP1 expression without any further validation studies. 

All the figures look pixelated and could be submitted with high resolution.

RESPONSE: The figures were revised and present a resolution of 

  1. Reviewer’s comment: Figure 1: NRP1 gene expression in different molecular subgroups of human MB 

- Would be better to put the groups in order (WNT, SHH, Group 3, Group 4) from left to right

RESPONSE: The order of the groups has been changed in the figures as suggested.

  1. Reviewer’s comment: Any comments on WNT subtype? Seems like it has high expression of NRP1 as well.

RESPONSE: That is an excellent suggestion. We have extended our comments on results from WNT tumors in the Discussion section.

  1. Reviewer’s comment: Says for values to be statistically significant, then p<0.001. Methods says p<0.01. 

RESPONSE: We would accept p < 0.01 as significant, although the results showed differences at the p < 0.001 level.  

  1. Reviewer’s comment: Figure 2: NRP1 gene expression across different subtypes of human MB 

- Some Group 4 and WNT subtypes also seem to have higher expression levels of NRP1 – would be helpful if researchers commented on this as well  

RESPONSE: We have included comments on the differences between subtypes within subgroups WNT and Group 4 in the Discussion section.

  1. Reviewer’s comment:

Figure 3:

- Paper comments on SHH, G3, and G4 patients. Should comment on WNT as well as there is a figure panel for it.

RESPONSE: A comment on the OS result for patients with Wnt tumors has been included in the Discussion section.

  1. Reviewer’s comment: Other comments: 

-       Researchers can go a step further and try to use other patient data sets to validate their findings 

RESPONSE: We have followed this suggestion. Additional analysis using a second tumor cohort, the Tumor Medulloblastoma – Pfister (n = 67), originally described in reference [5], confirmed our first data, showing significantly lower NRP1 transcription levels in Group 3 (n = 41, p < 0.001) and Group 4 (n = 64, p < 0.0001) tumors compared with SHH tumors (n = 46, Figure 4 in the revised manuscript).

  1. Reviewer’s comment: Perhaps suggest a reason for why SHH and G3 tumours with low NRP1 expression, have poor prognosis when the literature says otherwise (high NRP1 = worse prognosis in multiple cancer types) 

RESPONSE: This is an excellent suggestion. We have addressed this issue in the revised Discussion section, as follows: “Although most previously published findings indicate that NRP1 promotes tumor growth and contributes to reducing cancer patient survival [17, 21], it might not always the case. For instance, the NRP1 ligand PIGF acts as a pro-tumoral factor in some models, but can inhibit tumor growth in others [25]. Moreover, higher NRP1 expression was reported to be associated with a longer survival time in patients with neuroblastoma (NB), a pediatric tumor usually arising from the peripheral nervous system. The same study showed that NRP1 knockdown reduced migration and invasion in NB cells, suggesting a tumor suppressive role for NRP1, via activation of β1 integrin [26].”

  1. Reviewer’s comment: Must improve figures. They are blurry and not easy to read

-       What is the rationale for choosing NRP1? How did the researchers come about finding NRP1 as a gene target (apart from it being overexpressed in other cancers)? Why did they decide to look at it in specific subtypes?

RESPONSE: Given that pediatric brain tumors likely arise from abnormalities in CNS development during embryogenesis, they may be particularly sensitive to factors that influence neurodevelopment, such as NRP1. We have included this comment sin the revised Introduction to clarify the basis for our study’s rationale.

  1. Reviewer’s comment:

-       In the discussion, authors wrote “MRP1” instead of NRP1 

RESPONSE: This has been corrected in the revised manuscript.

  1. Reviewer’s comment: The authors have mentioned in methods that “MicroRNA ex-153 pression levels were normalized using the R2 Genomics Analysis and Visualization Plat-154 form (http://r2.amc.nl).” Where is this data displayed? 

RESPONSE: This mention to microRNA was included by mistake and has now been corrected.

  1. Reviewer’s comment: Although the finding that low expression of the NRP1is associated with shorter survival in patients with SHH and Group 3 Medulloblastoma seems noteworthy, interest to the readers may be dampened without any experimental validation studies. 

RESPONSE: Our article is a rapid communication highlighting a potential novel role for NRP1 expression as a predictor of survival in MB patients. Future experiments should examine this issue using experimental assays in MB cells.   

Round 2

Reviewer 2 Report

The revised manuscript has been improved.